# Androgens and Androgen Receptor Actions on Bone Health and Disease: From Androgen Deficiency to Androgen Therapy

**DOI:** 10.3390/cells8111318

**Published:** 2019-10-25

**Authors:** Jia-Feng Chen, Pei-Wen Lin, Yi-Ru Tsai, Yi-Chien Yang, Hong-Yo Kang

**Affiliations:** 1Division of Rheumatology, Allergy and Immunology, Department of Internal Medicine, Kaohsiung Chang-Gung Memorial Hospital and Chang Gung University, College of Medicine, Kaohsiung 833, Taiwan; uporchid@cgmh.org.tw; 2Graduate Institute of Clinical Medical Sciences, College of Medicine, Chang Gung University, Kaohsiung 833, Taiwan; peiwen349@gmail.com (P.-W.L.); sexhappiness@gmail.com (Y.-R.T.); yichienyang@gmail.com (Y.-C.Y.); 3Center for Menopause and Reproductive Medicine Research, Department of Obstetrics and Gynecology, Kaohsiung Chang-Gung Memorial Hospital and Chang Gung University, College of Medicine, Kaohsiung 833, Taiwan; 4An-Ten Obstetrics and Gynecology Clinic, Kaohsiung 802, Taiwan; 5Department of Dermatology, Kaohsiung Chang Gung Memorial Hospital, Kaohsiung 833, Taiwan

**Keywords:** androgens, androgen receptor, osteoporosis sex differences, and bone regeneration

## Abstract

Androgens are not only essential for bone development but for the maintenance of bone mass. Therefore, conditions with androgen deficiency, such as male hypogonadism, androgen-insensitive syndromes, and prostate cancer with androgen deprivation therapy are strongly associated with bone loss and increased fracture risk. Here we summarize the skeletal effects of androgens—androgen receptors (AR) actions based on in vitro and in vivo studies from animals and humans, and discuss bone loss due to androgens/AR deficiency to clarify the molecular basis for the anabolic action of androgens and AR in bone homeostasis and unravel the functions of androgen/AR signaling in healthy and disease states. Moreover, we provide evidence for the skeletal benefits of androgen therapy and elucidate why androgens are more beneficial than male sexual hormones, highlighting their therapeutic potential as osteoanabolic steroids in improving bone fracture repair. Finally, the application of selective androgen receptor modulators may provide new approaches for the treatment of osteoporosis and fractures as well as building stronger bones in diseases dependent on androgens/AR status.

## 1. Introduction

The discovery of androgen actions initiated in the 19th century with the idea of self-injection with testicular extracts from pigs or dogs to restore vitality by Dr. Charles Brown–Séquard [1]. It was known that the development and maintenance of male characteristics are modulated through androgen receptors (AR) [2], also designated as NR3C4 (nuclear receptor subfamily 3 group C member 4). Androgens bind to AR in the cytoplasm, translocating into the nucleus, and binding to DNA to function as a transcription factor. [2]. Androgens/AR signaling axis is a requisite for androgenic phenotype expression through regulating diverse responses at target tissues, including bones. The most important androgens, principally testosterone (T) and 5α-dihydrotestosterone (DHT), influence the human skeleton in both males and females. Testosterone exerts anabolic effects on the skeleton through signal transduction via (a) binding to AR, and (b) conversion to 17-beta estradiol (E2) via the enzyme aromatase (from cytochrome P450 family) which then binds to estrogen receptors (ERs) [3,4]. Conversely, both circulating and peripheral effects of testosterone on the body depend on its conversion to DHT by the local enzyme 5α-reductase (SRD5A; type I and II) in target tissues. Both T and DHT bind to AR, but DHT has a higher affinity to AR in many tissues. The activation of AR and ERα, but not ERβ, is associated with maintenance of the trabecular bone. The effects of ERα activation preserved the thickness and number of trabeculae, while AR preserved the number of trabeculae [5]. For cortical bone, the bone-sparing effects were majorly mediated by ERα, but not AR or ERβ for the maintenance of the cortical thickness, volumetric density, and mechanical strength [5]. Activation of ERα may have direct impacts on bone, or indirectly through increasing Insulin-like growth factor-1 (IGF-1) in serum [5].

## 2. Androgens and Androgen Receptors (AR) on Bone Health

### 2.1. The Effects of Androgen and Androgen Receptors on Bone Growth

Androgens are known to stimulate longitudinal bone growth as well as radial bone growth, thereby increasing the cortical bone size. The longitudinal bone grows through the endochondral bone formation and epiphyseal plate growth, whereas the radial bone grows through periosteal apposition. Cartilage cells, predominantly chondrocytes, proliferate and differentiate under the regulation of various endocrines, such as growth hormone (GH), insulin-like growth factor-I (IGF-I), transforming growth factor (TGF-beta), and vitamin D metabolites [6]. Longitudinal bone growth is governed by sex hormones that exert biphasic effects during adolescence—as puberty begins, androgen and estrogen stimulate endochondral bone development, but the epiphyseal growth plate closure is majorly mediated by estrogen via ER at the end of puberty through aromatization of androgen to estrogen [7]. Estrogen demonstrates biphasic effects on epiphyseal growth, where low concentrations, as male sex physiologically presented, can stimulate epiphyseal growth, whereas higher concentration level, as female sex presents, is associated with the arrest of bone growth [8]. This is testified by the observed growth spurt in puberty due to the delayed closure of epiphyses in estrogen-deficient individuals (e.g., mutated aromatase gene) or estrogen-resistant cases (ER gene mutation) [9,10]. 

A greater radial bone expansion, comprising enlargement of bone diameter and increase of cortical thickness, is characteristic for bone growth in male puberty. Accelerated periosteal bone apposition in men than in women [11] is conventionally supposed to result from the stimulatory effects of androgens on periosteal bone through AR in men and the inhibitory effects of estrogens in women during puberty. Experiments in transgenic mice provided evidence that estrogen may also stimulate radial bone growth. Observational studies on men with aromatase deficiency, characterized by normal serum androgen but undetectable levels of estrogen, exhibit low bone mass and areal density. Taking these findings together, periosteal bone expension is not only mediated by androgen, but also estrogen. 

Previous studies documented sex-related dimorphism in the cortical bone size, and strength in rodents and to a lesser extent, in humans [12]. While it is known that androgen actions enhance radial bone growth through increasing periosteal apposition in pubertal male rodents [13], their postpubertal effects at the periosteal surface are still unclear. Considering higher fracture risk in females and that volumetric bone density does not differ between the genders significantly, higher bone strength in males is likely a consequence of larger bones. In that context, it will be especially valuable to elucidate how bone size is regulated and which biochemical mechanisms drive periosteal apposition of bone material. It was suggested that androgen effects on radial bone growth are mediated both via AR and ER signaling pathways in male mice [14]; nevertheless, it is still unknown how these pathways are mutually related. Periosteal expansion is mediated via AR and ERα pathway [15]. However, the target cell(s) mediating these effects remain unclear. It was shown that a deficiency of estrogens by aromatase inhibitors impairs periosteal bone formation and endosteal bone resorption during radial bone growth in orchidectomized male rats [16]. The longitudinal bone growth is facilitated by chondrocytes at the epiphyseal plates, closure of which is mediated by estrogens on the chondrocytes [15]. This scenario might explain that the earlier growth arrest is attributed to the earlier onset of puberty in females. However, further studies are required to reveal molecular mechanisms behind these observations (e.g., direct or indirect mechanisms, growth hormone (GH)/IGF-1 axis activation at different time points and concentrations).

The GH/IGF-1 axis is essential in achieving a normal longitudinal bone growth and bone mass acquisition in the association with sex steroids. GH not only directly promotes differentiation of mesenchymal stem cells (MSC) into osteoblasts but stimulates osteoblast proliferation [17,18]. GH may promote osteoblast differentiation indirectly by upregulating bone morphogenetic proteins (BMPs) and IGF-1/IGF-2B [19]. IGF-1, majorly produced in the liver, exerts stimulatory effects through binding to IGF-1 receptors on osteoblasts resulting in cell differentiation [20], and the serum levels are positively correlated with BMD in older women and men [21]. Circulating estrogen, whether endogenous or exogenous in origin, may modulate the GH/IGF-1 axis, influencing bone turnover. In a GH receptor gene knockout (GHRKO) mouse study, GH/IGF-I activation is not involved in the development and maintenance of trabecular bone, while androgens stimulate trabecular bone modeling independently from GH/IGH-1. However, GH/IGF-1 activation are the major determinants of periosteal bone growth to obtain optimal periosteal bone growth [13].

### 2.2. The Effects of Androgen and Androgen Receptors on Bone Homeostasis 

Application of AR transgenic and AR knockout (ARKO) mouse models have recently shed more light on the AR role in androgen effects on bone homeostasis [22]. Previous studies showed different effects of androgens depending on the species, age, sex, and specific bone compartment; yet, it seems that irrespective of age or sex, they preserve trabecular bone integrity. Under the androgen effects, the trabecular bone may be maintained directly by osteocytes, or indirectly by inhibiting osteoclastogenesis through interaction with osteoblast precursors, while AR has no role on osteoclasts [23,24,25]. ARKO mice have shown significant bone deterioration already at a young age. Nevertheless, whether sex hormone actions influence bone homeostasis in aged male and female mice through the regulation of AR activity remains unclear. Recent studies have evaluated how sex hormone treatment influences bone microarchitecture in ARKO mice and showed that bone mass decreased in ARKO mice of both sexes at six, 18, and 30 weeks of age, and that AR deficiency has a greater effect on bone microstructure at six weeks in males. Gonadectomy did not further worsen the bone microstructure in ARKO mice at 18 weeks of age [26]. Bone strength and stiffness decreased in female and male ARKO mice, respectively. E2, but not DHT or dehydroepiandrosterone (DHEA) treatment rescued trabecular bone mass and bone stiffness in both sexes of gonadectomized ARKO mice [26]. In summary, the inactivation of AR affects bone remodeling in both male and female. Both trabecular and cortical bones are affected. However, the influences vary according to age, gender, and the type of supplemented hormones. Additional studies are required to unravel the detailed mechanisms of interactions in bone homeostasis. 

### 2.3. Regulation of Bone Turnover in Men

Bone turnover markers (BTM), products derived from bone matrix or bone cells during bone remodeling, could be categorized by either catabolic markers, such as C- or N-terminal telopeptide of type I collagen (CTX or NTX), pyridinolines, or anabolic markers, such as procollagen type I N-terminal propeptide (PINP) bone alkaline phosphatase (bone ALP), or osteocalcin (OC). For men, the bone turnover markers are highest in the second or third decade, then mildly decrease hereafter. In young men, the relationship between increased bone turnover and high levels of sex hormone or IGF-1 is stronger than that of old men [27]. Data from the Dubbo Osteoporosis Epidemiology Study, MINOS cohort study of osteoporosis in men, and the Osteoporotic Fractures in Men (MrOS) Study are unable to prove conventional BTM as an independent biomarker that could predict fragility fractures in older men [28,29,30].

AR is predominantly expressed in osteoblasts and osteocytes, but not on osteoclasts [24,31,32]. It has been shown that osteoclast-specific ARKO mice have no changes in osteoclast surface, bone microarchitecture, or the response to orchidectomy and androgen replacement, suggesting that the AR in osteoclasts has no critical role for bone maintenance in mice [31]. Regulation of osteoclast function is primarily mediated by estrogens and ERs, since T may control osteoclast indirectly by aromatization to E2, or by inhibiting the production of interleukin-6, which is essential for the maturation of osteoclast precursors [33]. AR in osteoblasts is up-regulated by androgen, estrogen, and 1,25-dihydroxyvitamin D to stimulate osteoblast proliferation, differentiation, and synthesis of extracellular matrix protein and mineralization [34,35]. AR in osteocytes had a direct role in maintaining skeletal integrity and bone quality [24]. However, AR had no role in the direct action on osteoclasts in animal model studies [31]. 

## 3. The Impacts of Androgen Deficiency/Excess on Bone Remodeling in Human

### 3.1. Hypogonadism in Men (Androgen Deficiency)

Androgen deficiency manifested as a delay in sexual development, reduced testicular volume, axillary and pubic hair loss, and other suggestive symptoms included reduced libido, erectile dysfunction, gynecomastia, low sperm count and skeletal disorders, such as height loss, low traumatic fracture, or decreased BMD [36]. To diagnose hypogonadism in men, apart from the above-listed symptoms and signs, the patient should show consistently low T levels (total T and free T levels in serum) [36]. Bone mineral density (BMD) is notably lower in hypogonadal men (especially at the spine—a predominantly cancellous bone site) than in their healthy counterparts [37]. Impaired skeletal growth with lower peak bone mass achieved reflects in decreased BMD at the spine, femur, and distal sites of non-dominant radial bone [38,39]. Moreover, it was reported recently that hypogonadism is associated with a deteriorated trabecular microarchitecture of the distal tibia [40]. The significant relationship between pretreatment serum T levels and BMD was found in congenital hypogonadal men [41]. One large prospective, multicenter-based cohort, MrOS study, followed 2908 men, discovered that free testosterone was a positive predictor for BMD in the total body, total hip, and femur trochanter. Also, a low T level was associated with an increased number of fragility fractures, including vertebral fracture [42]. Other studies in men with hip or vertebral fractures highlighted that a number of individuals were hypogonadal [43,44]. T supplementary therapy has also been proved to contribute to BMD increment in men with low testosterone levels in several studies [41,45,46,47,48,49,50]. 

While some studies in adult hypogonadal men showed a major increase in bone resorption and a lesser increase in bone formation, similar to postmenopausal women [51], others emphasized low bone formation in these individuals [52,53]. The previous studies with a histological survey of follow-up biopsies supported the concept that T stimulates bone formation and mineralization [53]. A reduction in plasma 1,25-dihydroxyvitamin D (1,25(OH)2D) was found to be a major risk factor for osteoporosis in hypogonadal men [52]. Following T supplementary therapy significantly increases in 1,25(OH)2D/ calcium absorption [40] and serum bone formation markers [42], implicating that androgen is contributory to bone formation. In Table 1, we summarized different clinical conditions associated with androgen deficiency or excess, and discuss them seperately.

### 3.2. Isolated Hypogonadotropic Hypogonadism (IHH)

In contrast to other types of male hypogonadotropic hypogonadism, IHH is an early and complete form of hypogonadism with isolated sex steroid deficiency and no other metabolic abnormalities, which makes it a good model to examine the effects of sex steroids and sex steroid deficiencies in men. Compared with age-matched controls, patients with IHH have lower bone density at the spine and radius, not only before but also after growth plate closure [71]. Interestingly, both areal and volumetric bone density are reduced. This suggests that the exact bone composition is impaired in the context of IHH. In these patients, assessing bone turnover has produced inconsistent results, demonstrating histomorphometric evidence for low-turnover osteoporosis in some patients [72] but increased levels of markers of bone formation and resorption in others [54].

There is also data about lower bone density (at radius and spine) in hyperprolactinemic hypogonadal men [54,73]. Nevertheless, in these men, cortical bone density could be improved by T supplementation, regardless of the prolactin levels, emphasizing hypogonadism as the primary factor for impaired bone homeostasis in these cases [74].

### 3.3. Klinefelter’s Syndrome (KS)

Klinefelter’s syndrome (KS), first reported in 1942 by Klinefelter et al., represents the presence of an extra X chromosome in a male karyotype, resulting in a typical phenotype which includes tall stature, small testes, aspermatogenesis, gynecomastia, diminished body hair, and other signs of androgen deficiency [75]. KS is recognized as the most common form of hypergonadotropic hypogonadism resulting in functional and structural testicular insufficiency during puberty. Impaired bone metabolism begins in early life, and children and adolescents with this syndrome could exhibit impaired bone structure as reflected in altered ultrasound parameters of phalanges, significantly lower serum 1,25(OH)2D level, markers of bone formation, and higher parathyroid hormone (PTH) level [76]. The long-term consequence of adult KS demonstrated universally lowered BMD at the spine, hip, and forearm than the age-matched healthy controls, as well as lower 1,25(OH)2D level. However, T is not an independent predictor for BMD [56]. KS patients are prone to having higher risks of fracture leading to morbidities and mortalities [77,78,79]. Although a positive effect on bone density was reported if testosterone treatment was initiated before the age of 20 years [80], in a three-year-followup study, T therapy in KS only increased lumbar spine BMD, but failed to improve parameters evaluated based on the trabecular bone score (TBS) or hip structure analysis (HSA) [81]. Interestingly, even patients on long-term T supplementation still showed low BMD [82,83], and several studies emphasized ill correlation of the serum T level and bone parameters [56,84]. This observation raises the question whether androgen deficiency is the main reason for the low BMD and whether T replacement should be the primary treatment choice to improve BMD in this specific group of patients; moreover, it suggests that bone deterioration in these patients may be, at least in part, independent of the actual hypogonadism. These results may also reflect the fact that KS patients display varying degrees of hypogonadism, where only severe hypogonadism may be linked to bone loss [85,86]. Considering the methodological limitations of the majority of previous studies (small sample size, lack of control group), large prospective and controlled trials are needed to elucidate how the bone structure is influenced by different levels of hypogonadism and T replacement.

### 3.4. Constitutional Delay of Growth and Puberty (CDGP)

Transient hypogonadotropic hypogonadism or constitutional delay of growth and puberty (CDGP) is the most common cause of delayed puberty, characterized by delayed skeletal age and a family history of delayed growth and puberty. Compared to healthy children, boys with CDGP have normal bone turnover markers, the BMD increases in a similar manner [87]. In boys with CDGP, androgen therapy proved useful to improve bone mass [88] and calcium absorption/retention in bone [89]. In clinical studies, androgen treatment contributes to growth velocity only, but it does not affect the gain of adult height [90,91,92]. Therefore, considering the high likelihood of the spontaneous restoration of normal pubertal development and attainment of their mid-parental target height, it is acceptable that observation and reassurance are a reasonable strategy [90].

### 3.5. Androgen Deprivation Therapy and Castration

It was reported that in men with advanced prostate cancer, androgen deprivation therapy (ADT) leads to accelerated bone degradation and increased risk to sustain a bone fracture [93,94], which resembles bone deterioration occurring in ovariectomized or early postmenopausal women. A meta-analysis in prostate cancer patients who underwent ADT showed a significant decline of BMD within three years follow-up, with the mean decline of 3.7%, 3.1%, and 1.6% in the regions of the lumbar spine, femoral neck, and total hip, respectively; these rates were faster than the yearly bone loss rate, around 0.5–1.0% in the normal middle-aged men [95]. Nevertheless, it should be noted that many prostate cancer patients could already have decreased BMD even before ADT was started [96]. A long-term prospective study following five years enrolled more than 50,000 men with prostatic cancer, discovering a significantly higher fracture rate developed in ADT patients compared to non-ADT after adjusting confounding factors (19.4% in ADT vs. 12.6% in non-ADT, *p* < 0.001) [97]. Non-metastatic prostate cancer patients who underwent ADT without anti-resorptive therapy may suffer from a 21–37% increase in fracture risk [97,98,99]. Fracture risk in these patients may further increase due to the decreased lean mass secondary to androgen deficiency [100,101,102].

Based on these findings, there is much resemblance in bone deterioration between androgen-deficient men and postmenopausal women or hypogonadal animals. In this sense, imbalances in bone remodeling with a shift to resorption would cause net bone loss, which is evident at intracortical or endocortical surfaces in the appendicular skeleton and vertebral bone [103,104]. That would signify that sex steroid deficiency in men leads to high-turnover osteoporosis, too. Despite the lack of direct microstructural confirmation for this notion, clinical experience clearly shows that bisphosphonates (etidronate [105], i.v. pamidronate [106], zoledronic acid [107])—as turnover suppressors could prevent bone loss in those patients and may represent a sound therapeutic approach after castration.

### 3.6. Aging

Circulating T is transported predominantly by sex hormone-binding globulin (SHBG), which controls the amount of T in the body. SHBG-bound T is biologically inactive due to high affinity, whereas free T (approximately 2–3%) [108] and albumin-bound T are considered bioavailable, and correlate better with BMD and muscle mass than total T [109,110]. In aging men, total T declined at a speed of 1.6% per year, while 2–3% per year for the bioavailable T [111]. However, SHGB level increases with aging, resulting in a greater reduction of bioavailable T level [111]. Total and free Serum T inversely correlated with age and are associated with sexual symptoms, physical activity, and metabolic conditions in different cohorts [112,113,114]. The reference range of T, influenced by races, regions, timing of sample collection, and laboratory methods, need to be harmonized to avoid inter-cohort variation [115].

Aging in men results in a progressive reduction of hypothalamic-pituitary-gonadal (HPG) axis function, decreasing testosterone secretion through both central and peripheral origins. Data from the Osteoporotic Fractures in Men Study (MrOS) indicated that sex hormone deficiency was associated with higher prevalence of osteoporosis at baseline and greater loss of BMD over time in old age men accompanied with parallel T and E2 decline, of which 3% were T deficient, 3.2% were E2 deficient, and 0.7% were deficient in both. In old age men, there was a threshold level of E2 for fracture [116] but no association between total T and fracture, implicating that E2, but not T, may be the major sex hormone associated with fracture risk in older men. On the contrary, some studies indicated that serum T is associated with fractures independently [42] and more powerfully than E2 [117]. For instance, high bone resorption in men who sustained a hip fracture correlated well with low serum T levels [118]. However, old age and sex hormone deficiency might be two distinct risk factors of cortical bone loss in old age from an animal study, which addressed the major mechanism of estrogen deficiency in bone through an increase osteoclastogenesis, whereas aging was through a decrease in osteoblastogenesis in combination with an increase osteoclastogenesis [119].

In contrast, there are scarce proofs of the beneficial impact of androgens in postmenopausal women. It is a fact that adrenal androgens (e.g., DHEA-S) serum levels drop by 70% in menopause) [120,121], but we do not know how that impacts bone loss. While there were no prospective studies on the relationship with fracture risk, previous cross-sectional studies failed to find any consistent relationship between serum DHEA-S and BMD [122,123]. Further studies are also needed to clarify whether DHEA-S has direct androgen/AR-mediated effects in the skeleton, or merely serves as a source for aromatization into estrogens.

### 3.7. Androgen Insensitivity Syndrome (AIS)

Androgen insensitivity syndrome (AIS) is caused by a mutation in the gene encoding the AR (Xq11–q12) [124]. AIS affects individuals with a 46, XY karyotype and its presentation usually involves the presence of feminized external genitalia at birth, abnormal development of secondary sexual traits in puberty, and infertility [124,125]. According to the range of defects in androgen action and appearance of external genitalia, AIS is subcategorized as complete form of AIS (CAIS) with female genitalia, partial androgen insensitivity syndrome (PAIS) with ambiguous genitalia, or with the predomination of either sex, and the mild form (mild androgen insensitivity syndrome, MAIS) in which external genitalia are typically males [125]. The therapeutic strategy relies on the reinforcement of the sexual identity, as well as planning the gonadectomy and hormone replacement therapy.

Adult CAIS women showed decreased BMD values [126,127] both before and after gonadectomy due to a combination of skeletal resistance to androgen action and estrogen deficiency [128]. Although reduced BMD may stem from the impaired function of AR, there is still a debate on the association between gonadectomy and BMD [126,127]. AR signaling seems to influence trabecular bone preferentially [65,126]. In AIS women with intact gonads, BMD loss predominantly developed at the lumbar spine rather than the femoral neck [65,129,130]. One previous study revealed that the femoral neck in CAIS with intact gonads was within the normal range [127]. As shown in animal models, AR-dependent maintenance of cancellous bone targets osteoblasts and osteocytes, the leading players of matrix synthesis and mineralization [24,127,131,132].

A reduction in the lumbar and femoral neck BMD was reported in AIS women who underwent gonadectomy [127,128,133], and BMD reduction at the lumbar spine is more profound than that at the femoral neck [65,128,134]. However, these studies may enroll participants affected by 46, XY sex disorders other than AIS [127,134]. It is well established that the decline in BMD can be regarded as a major determinant of fracture risk [135], but the fracture rate of AIS was rarely investigated from previous studies and reviews. Only one study reported a high fracture rate (27.3%) in women with removed gonads [134]. Since decreased BMD could be found in some cases with intact gonads, it was suggested that supplementing with additional estrogens could be helpful for them even if they demonstrate intact testes [136]. It is also essential to optimize doses/formulations of hormone replacement therapy in CAIS women following gonadectomy, preferably using transdermal estrogen patches and strictly monitoring compliance.

### 3.8. Hyperandrogenism (Androgen Excessive)

Numerous studies showed an increased BMD in hyperandrogenemia, such as in women with polycystic ovary syndrome (PCOS) [25,137,138]. In contrast, some studies demonstrated lower BMD in PCOS compared to controls, although the serum T concentration was higher than the controls [68,139]. No difference in BMD and postmenopausal fracture incidence between PCOS patients and healthy controls have been reported [67,140]. These conflict results may imply that in different study groups, the variable degree of the excess of androgen production in PCOS may partially preserve the BMD but did not always overcome the deleterious bone effects of estrogen deficiency. Also, the effect of hyperandrogenism may be differential across the various bone sites according to different androgen [141,142]. PCOS model of androgen excess has a number of limitations, such as confounding factors that are difficult to control in case-control study design (differences in body mass index (BMI), body composition, and menstrual irregularities), and often the crude determination of androgen excess by the presence of hirsutism. Nevertheless, PCOS studies contributed to the understanding of the link between hirsutism and peak BMD (e.g., compared to age-matched controls, hirsute women showed higher peak BMD, even after adjustment for BMI) [143,144]. Owing to the findings from the peripheral quantitative computed tomography (pQCT) study [138], increased BMD reflects real changes in bone composition, not just the changes in bone size [141]. However, it is still unknown if these positive effects are mediated through AR or rely on aromatization. It was observed that PCOS patients with high BMI tend to have higher BMD than those with normal or low BMI, signifying the importance of aromatization within fat tissue [145]. Additional research is warranted to clarify potential skeletal effects of other metabolic abnormalities in PCOS (such as low levels of SHBG with increased hormone bioavailability, or high insulin level) [145]. Insulin appears to be one of the most crucial positive bone growth stimulators through direct and indirect effects on bone formation, and BMD is a significant negative correlation with insulin sensitivity index [140,146]. Insulin resistance and hyperinsulinemia may directly stimulate production of androgens via ovarian tissue steroidogenesis by inhibiting hepatic sex hormone-binding globulin (SHBG) production, thus increasing free testosterone [147,148,149].

Some of the treatment options used in hirsute women with androgen excess (namely, gonadotropin-releasing hormone (GnRH) agonists and AR antagonists, alone or together) may have adverse effects on bone metabolism. For instance, GnRH agonist therapy would suppress estrogen and androgen levels and thereby facilitate bone loss in hirsute women, although the effect is preventable [150]. While in a six-month period, the use of spironolactone as an AR antagonist was able to preserve BMD in hirsute women on GnRH agonists [151], spironolactone was usually associated with a reduced BMD [152]. On the other hand, another AR antagonist flutamide, when administered alone, did not affect BMD values of the lumbar spine [153] Metformin treatment was associated with reduced bone turnover, as suggested by reductions in markers of bone formation and resorption, leading to slower bone remodeling in premenopausal women with PCOS[154]. However, controversy still exists, and these findings should be taken with caution as they were based on uncontrolled, short-term studies. Adjustment for differences in age, duration of amenorrhoea, lifestyle factors, differences in body composition, or weight distribution, BMI, and various ethnic groups could provide a better understanding of the relation between bone and PCOS.

## 4. Androgens/AR Actions on Skeletons in Animal Studies

Female mouse ovariectomy (OVX)-induced surgical castration, and male mouse orchiectomy (ORX) represent the most commonly used surgical castration method for studying the effects of skeletal sex steroids. Cortical bone growth was also observed in male ORX mice, as described in rats [155,156,157]. Female mice did not demonstrate remarkable periosteal expansion after ovariectomy compared with rats [158,159]. In cancellous bone, both OVX [158,159,160,161] and ORX [33,156,157,160,162] led to a loss of bone mass, that was associated with raised bone turnover markers [156,158], and thus OVX and ORX mice are increasingly considered to be animal models of steroid action studies in postmenopausal-like osteoporosis [15,25]. Besides, DHT was able to reverse the loss of cancellous bone, but DHT did not prevent cortical thinning induced by ORX [163]. In summary, both DHT and T in OVX female and ORX male rats were protective for bone integrity, in particular in the cancellous bone. DHT does not appear to be as effective as T in cortical bones, maybe because T can undergo aromatization through estrogens and stimulation of the ERs.

Testosterone supplementation can prevent bone loss effectively in ORX mice cancellous bone [33,156], and DHT [164] and estrogens [155,156,159,160,162,164] also seem to help preserve bone after gonadectomy in both sexes. Phytoestrogens have similar bone-preservation effects observed in OVX [165] and ORX mice [166]. Therefore, loss of cancellous bone in mice, regardless of sex, could be prevented by both androgens and estrogens. Also, T increases the cortical region of ORX mice [5,167], and delivery of T with biodegradable stents is effective in promoting bone healing in femur fractures of mice with critical size segmental defects. [168].

Other animal studies involved animals with knockouts for AR and ERs. Those studies mainly revealed the roles taken by AR and ERs to manipulate the genetic conditions of male animals. For instance, it showed that ER rather than AR activation involves longitudinal bone growth [169]. In addition, knockout mice have shown that both ERα and AR are involved in enhancing cortical radial bone growth. Unlike AR and ERα, ERβ cannot mediate retention of sex steroids in the cancellous bone [162,170]. Furthermore, two modes of action were suggested for T: a direct action via AR and indirect action via ERα through aromatization [171].

Transgenic expressed AR transgenic mice driven by a 3.6 kb fragment showed that AR was overexpressed in all cells of osteoblast lineage [172,173,174], and expressed a phenotype of reduced turnover leading to reduced bone strength [175]. Unlike in wild type mice, DHT treatment was not able to increase cortical thickness and restore bone volume in AR3.6 transgenic mice [176]. Microstructural analysis of cancellous bone by μCT revealed significant loss of bone volume and trabecular number after ORX. Immediate DHT replacement prevented degradation of trabecular bone, particularly in AR transgenic mice, as compared to controls [23,176], suggesting enhanced androgen responsiveness in osteoblasts and osteocytes. Ablating mature osteoblasts’ AR decreased bone mass and reduced the number of trabeculae, indicating a shift towards bone resorption [23,177]. The decline in trabecular number after ORX in AR transgenic animals was prevented in the state of high turnover, where DHT treatment even increasing the trabecular number [176]. These data show the importance of supporting androgen signaling to reduce bone turnover via AR, probably by inhibiting osteoclasts formation or activity via osteocyte-mediated effects.

However, additional knowledge on the roles of AR and ER in bone metabolism stems from research applying cell-specific deletions using the Cre/LoxP technique. Cre/LoxP recombination is a site-specific recombinase (Cre enzymes) technique. Cre recombinase recombines a pair of short target sequences called the LoxP sequences. The technique used to carry out deletions at specific sites. In this context, the global deletion of AR in male mice led to significant alterations, particularly in serum testosterone concentration [22,178]. Therefore, although the volume of cancellous bone and cortical bone of male AR knockout mice is reduced due to an increase in bone resorption, it remains to be clarified if it occurred due to AR loss or rather due to the simultaneous hypogonadism [179]. ARKO mice had reduced bone mass and decreased stiffness, and the effect of AR loss on the bone microstructure of male mice was greater than female mice [26].

While AR deletion in the cells of myeloid/osteoclast lineage did not lead to skeletal abnormalities, the deletion of the AR in osteoprogenitor cells (Prx1(paired-related homeobox gene)-Cre) leads to cancellous bone loss with higher resorption, the effects being more pronounced in males compared to female mice [23]. The findings were similar across Cre models in osteoblast lineage (e.g., Col1a1 (type I collagen)-, Ocn(osteocalcin)-, Dmp1(dentin matrix protein 1)-Cre mice crossed with floxed AR mice) [24,177,180]. Furthermore, in cancellous bone, these definitive effects of AR signaling in male mice seem inconsistent with the aforementioned human studies that estrogen is a major regulator of bone metabolism in men [179].

## 5. Molecular Mechanisms of AR Signaling in Skeletal Stem and Progenitor Cells

In earlier decades, the therapeutic potential of bone marrow mesenchymal stem/progenitor cells (BMSCs) was recognized, considering their ability to differentiate into osteoblasts and chondroblasts [181,182]. In this sense, BMSC transplantation was able to enhance the formation of new bone through promoted differentiation to osteoblasts and chondrocytes to improve bone-related diseases [183].

It is reported that androgen can enhance osteoblast differentiation and regulate the production and organization of bone matrix, improve the synthesis of extracellular matrix proteins, and stimulate mineralization [35]. Colvard et al. [184] discovered AR expression in cultured osteoblasts. Since then, AR expression was confirmed both in vivo and in vitro in osteoblasts and osteocytes [184,185]. Furthermore, in vitro studies showed AR binding in other bone cell lines (human osteosarcoma, osteoblast-like, and primary osteoblasts from several species) [32,186,187,188,189,190,191,192,193,194]. In humans, higher AR expression was found in osteoblasts from cortical bone compared to cancellous bone, with no sex-related differences [32,195]. Most studies [187,189,196,197], with some exceptions [32,198], indicated the ability of androgen to up-regulate its own receptor’s expression in osteoblasts. In the literature, the relative AR expression was examined during cultured osteoblast differentiation, demonstrating an increase in AR levels from proliferation through differentiation to reach maximum expression in mature mineralizing cultures [179,199].

There is sufficient evidence that osteoblasts are derived from bone marrow pluripotent mesenchymal stem cells. Some articles reported that bone marrow-derived mesenchymal stem cells (BMSCs) express the AR [200,201,202]. Also, the AR has been detected on megakaryocytes [201,203] and endothelial cells [201,203,204] in the bone compartment. Therefore, mediation of androgen effects on bone may involve other AR expressing cells, apart from osteoblasts and chondrocytes.

Considering BMSCs ability to differentiate into osteoblast lineage cells, they were in focus of investigations of AR signaling. ARKO mice show a reduction in bone formation by effects on osteoblast differentiation and mineralization [31,205]. Indeed, BMSCs had lower expression of osteoblast differentiation-related class genes in ARKO compared to wild type mice, as evidenced in microarray analysis [206]. Our previous study has shown that androgen therapy promotes two important indications for pre-osteoblast differentiation in MC3T3-E1 osteoblast precursor cells (alkaline phosphatase activity and calcium deposition), suggesting that androgen therapy on osteoblast differentiation has a positive effect [207,208]. Mechanistically, we identified several target genes needed for bone formation mediated by AR, such as TNSALP (the tissue-nonspecific alkaline phosphatase)-encoded alkaline phosphatase 2 (*Akp2*) gene, SIBLING (the small integrin-binding ligand N-linked glycoprotein) gene family [207,209]. Androgen/AR could induce *TNSALP* and *SIBLING* gene family expression by binding to the androgen response elements (ARE) motif in their promoter regions [207]. The androgen-enhanced *TNSALP* expression and activity lead to β-glycerophosphate hydrolysis increases inorganic phosphate (Pi) levels. The transport of Pi into the cell results in further stimulation AR expression and mineralization in osteoblasts, which could be inhibited by Pi transporter inhibitors or levamisole, a tissue non-specific alkaline phosphatase (TNAP) activity inhibitor [207]. The regulation of *TNSALP* and *SIBLING* gene expression by androgen/AR ensures that sufficient TNSALP activity is obtained on the outer surface of osteoblasts and matrix vesicles to provide sufficient Pi, and SIBLING matrix proteins required for the initial formation of hydroxyapatite crystals during osteoblastic mineralization (Figure 1).

Early studies have shown that androgen and AR signals inhibit embryonic stem cells’ self-renewal process [210,211]. Similar results were obtained in ARKO mice, indicating that AR deficiency enhances BMSC self-renewal [174,212,213,214]. Interestingly, it is still not sufficiently understood whether sex affects the self-renewal potential of these stem cells. For example, gender had little effect on self-renewal of BMSCs in one study [179,215], whereas another study demonstrated that the male gender had an inhibitory effect. The production of adipocyte-derived stromal cells (ADSCs) has been shown to have similar characteristics to BMSCs [216]. Importantly, better potential for self-renewal was ascertained in ADSCs and BMSCs isolated from ARKO compared to WT mice [202]. Further mechanistic studies indicate that BMSCs proliferation is promoted in AR deficiency through the activation of AKT and extracellular signal-regulated kinase (Erk)1/2 via increasing the expression of epidermal growth factor receptor (*EGFR*) [213]. Overall, all evidence suggests an inhibitory potential of androgen/AR signaling for BMSCs self-renewal by indirectly regulating AKT and Erk1/2.

There is a gender difference in adipose accumulation, and it is observed that male adipose tissue is less than female [217,218], indicating that androgen and AR signaling can restrain adipogenesis by inhibiting the function of pre-adipocytes and BMSCs. Here we proposed several molecular pathways of adipogenesis and osteogenesis to be regulated by androgen and AR signaling (Figure 2). Androgens and AR signaling increase expressions of insulin-like growth factor binding protein 3 (IGFBP3), thereby inhibiting the binding of insulin-like growth factor (IGF) to its receptor (IGF-R) [206,219]. IGF-R was required for adipocyte differentiation, and inhibition of IGF-R by androgen treatment results in less adipogenesis [206]. On the other hand, androgen treatments could inhibit expressions of adipogenic markers, such as peroxisome proliferator-activated receptor-gamma 2 (PPARγ2), lipoprotein lipase (LPL) and adipocyte protein 2 (aP2), likely through forming AR, β-catenin, and T cell factor 4 protein complex [206,220]. Together, these findings highlight that androgens/AR can modulate the functions of BMSCs to inhibit adipogenesis and promote osteogenesis.

## 6. Effects of Androgen Therapy on Bone

T replacement therapy (TRT) is recommended in hypogonadal men for the maintenance of sex characteristics and corrects androgen deficiency, although prostate cancer risk should be considered in different ages and populations [36]. Hypogonadism men, who are characterized by increased bone turnover and bone loss [100], would also benefit from androgen replacement therapy through suppressing bone resorption [53]. Likewise, androgens in hirsute women preserve bone structure even with low estradiol levels [143]; similarly, bone loss in postmenopausal women could be prevented by androgen therapy [221,222,223,224,225].

Regarding the efficacy of TRT on BMD, the results of data are conflicting. A meta-analysis collecting eight placebo-controlled trials including 365 participants demonstrated that TRT via intramuscular injection was associated with a gain in lumbar BMD, but a non-significant increase in the femoral neck, besides, T therapy via transdermal route had no impacts in this study [226]. T-undecanoate (TU), a long-acting TRT, significantly increases lumbar and femoral neck’s BMD after 36 months follow-up in middle-aged, hypogonadal men [48]. In an eight-year-follow up cohort study revealed similar positive effects of TU on the vertebral and femoral BMD [227]. For men with late-onset hypogonadism, TRT resulted in improvement of muscle mass, strength, and BMD, but data about the improvement in fall and fracture remained to be investigated.

Since the positive effect of T supplement on promoting bone health is not entirely via its effectiveness, considering that T could be aromatized to estrogen, which is also a contributor to bone metabolism. It is not clear how much of the T anabolic effects on bone in older men are from pure testosterone effects and how much derive from aromatization. In an attempt to unravel the relative effects of testosterone vs. estrogen, a study was conducted in which the participators were administered a long-acting GnRH agonist and an aromatase inhibitor. The changes of bone formation and resorption markers followed physiologic T and E replacement revealed the role of estrogen as the dominant regulator of bone resorption, while bone formation was maintained by both estrogen and testosterone [228].

Selective androgen receptor modulators (SARMs), similar to selective estrogen receptor modulators (SERMs) or other tissue-selective receptors binding molecules, activate the AR receptors in a tissue-specific manner without undesirable side effects, providing treatment of choice for various diseases, such as osteoporosis, breast cancer or muscular disorders [229]. The arylpropionamide-derived SARM, S-040503, increased BMD and the biomechanical strength of the femoral cortical bone in an animal model with orchiectomized rats, and similar results were also found in postmenopausal osteoporosis animal models with ovariectomized rats [230,231]. Another SARM in clinical trials, Enobosarm (GTx-024; GTx, Memphis, TN, USA), might lead to improvement of lean body mass and muscle wasting in cancer cachexia patients [232,233], but the effects on BMD or fracture need to be further investigated. The agonistic effects of SARM on bone, endothelium, and brain, and antagonistic effects on the prostate, might be equally significant, like SERMs in women. Considering importance of ER in cancellous bone, even older men with osteoporosis could benefit from the administration of SERMs with the appropriate tissue selectivity, while on the other hand, osteoporotic women may find SARMs equally helpful. However, we still need to reach a better understanding of how SARMs modulate the AR signaling and its impact on bone structure and metabolism. In this context, new insights may soon become available through clinical trials, application of new technologies of genetics, and proteomics along with the use of cell-specific transgenic and knockout mice.

## 7. Conclusion and Future Perspectives

There has been a lot of progress in our understanding of estrogen effects on bone, which already led to improved osteoporosis treatment. However, androgen actions on bone remain mostly unknown. Moreover, additional questions are raised through recent discoveries in the field of bone biology and physiology. It is of paramount interest to ensure further progress in the relatively unexplored area and unravel the mechanisms by which androgens affect bone formation and cellular activities.

In the past few years, we realized that androgens are important and required for skeletal health both in men and women, which forced us to rethink the actions of sex steroids in the skeletogenic stem progenitor cells within bone microenvironment and searched the potential treatment for bone injury such as segmental bone defects. The observation that androgens have anabolic effects in bone fracture repair in animals is also relevant for humans and might even help further optimize fracture healing in a clinical context. Based on the observed tissue specificity of SARMs, with a mixture of agonistic and antagonistic effects, the concept of androgens/AR targeted therapy against prostate cancer evolved; the subsequent and ongoing discoveries related to androgen signaling in osteoporosis and bone fracture indicate the potential for improvement of the skeletal stem progenitor cell activities. The ultimate intention of finding an “ideal” androgen that would provide regeneration potential against the major diseases in the bone may become a possibility in the near future.

## Figures and Tables

**Figure 1 cells-08-01318-f001:**
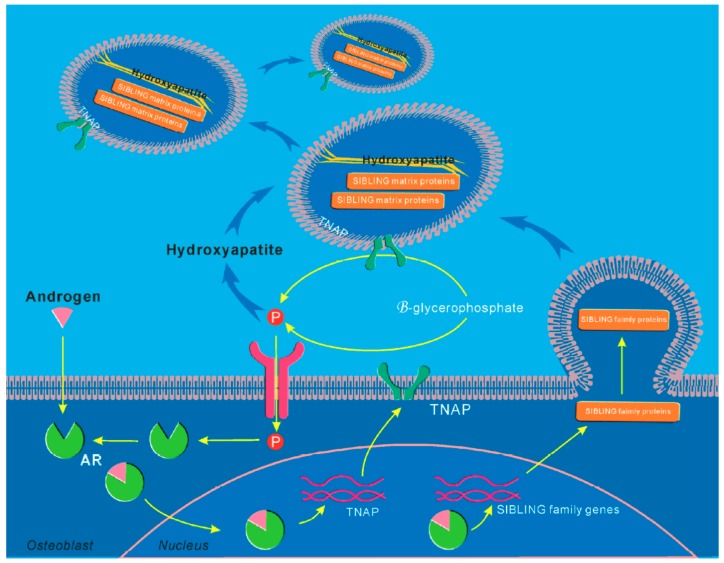
Molecular schemas of androgen/AR actions on osteoblastic mineralization. Androgen/AR induces *TNAP* and *SIBLING* family gene expression in osteoblasts by binding to the ARE motif in these gene promoter regions. The androgen-enhanced *TNAP* expression and activity results in an increased Pi concentration from the hydrolysis of β-glycerophosphate. The transport of Pi into the cell results in a further stimulation AR expression and mineralization in osteoblasts. The regulation of *TNAP* and *SIBLING* gene expression by androgen and AR ensures that sufficient TNAP activity is available at the outer membrane surface of osteoblasts and matrix vesicles to provide sufficient Pi, and SIBLING matrix proteins required for the initial formation of hydroxyapatite crystals during osteoblastic mineralization. ARE: androgen response elements; TNAP: tissue non-specific alkaline phosphatase; SIBLING: the small integrin-binding ligand N-linked glycoprotein

**Figure 2 cells-08-01318-f002:**
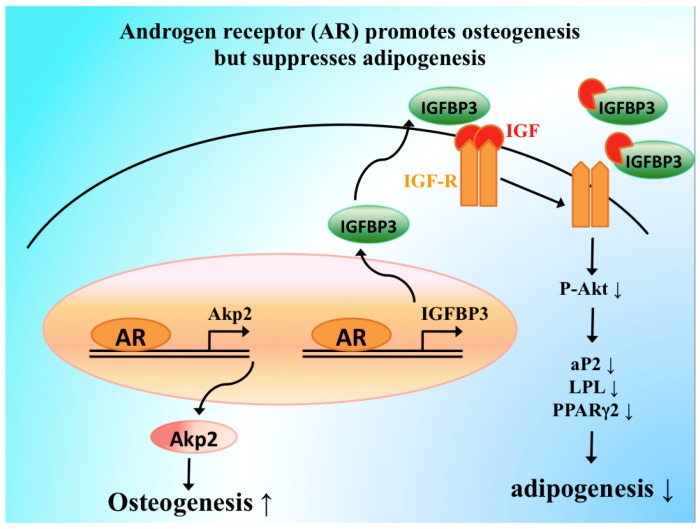
Androgen receptor suppresses adipogenesis but promotes osteogenesis in bone marrow stromal cells. AR increases insulin-like growth factor binding protein 3 (IGFBP3) through direct binding to its promoter region. The IGFBP3 increased by AR then participates in blocking insulin growth factor (IGF) receptor signaling to inhibit AK straining transforming (Akt) activation that leads to a decrease in the expressions of adipogenic markers, such as peroxisome proliferator-activated receptor-gamma 2 (PPARγ2), lipoprotein lipase (LPL), and adipocyte protein 2 (aP2), and causes less adipocyte differentiation. On the other hand, AR induces the expression of alkaline phosphatase 2 (*Akp2*) to promote osteogenesis.

**Table 1 cells-08-01318-t001:** Summary of impacts on bone in different disease associated with androgen deficiency or excess.

	Clinical Manifestation	Impacts on Bone in Adults	Reference
**Androgen Deficiency**			
Isolated hypogonadotropic hypogonadism (IHH)	Delayed puberty in late teens or early twenties.	lower lumbar spine, femoral neck, trochanter, and radius BMD. No data for possible increase fracture risk.	[54]
Klinefelter’s syndrome (KS)	tall stature, small testes, aspermatogenesis, gynecomastia, diminished body hair	Lower spine, hip, and forearm BMD in adult KS Increase fracture risk at the femoral area and mortality rate associated with hip fracture is 39%.	[55,56]
Constitutional delay of growth and puberty (CDGP)	short stature, delay bone age, and puberty	Failed to reach target height or predicted adult height. Significant increase of BMD in stage III and IV of puberty stages, and normal lumbar and femoral neck Volumetric BMD in adults. No increased risk for fracture.	[57,58,59]
Androgen deprivation therapy	Flushing, decrease libido, anemia, insulin resistance.	In prostate cancer patients, BMD decreased 3.7% at the lumbar spine and 2.1% at the femoral neck within the 1^st^ year. 2.7–8.1 % fracture within five years.	[60,61,62]
Aging	Degeneration of systemic change, sleep disturbance, decrease libido.	In women before menopause, an annual reduction rate < 0.4% and an increase to 1.2% after menopause. In men, continuous bone loss in the hip after peak bone mass after 50 years	[63]
Androgen insensitivity syndrome (AIS)	46,XY karyotype , with under masculinized external genitalia depends on residual AR function. Gynecomastia at puberty and infertility in adulthood	Normal pubertal growth for females in CAIS, appropriate epiphyseal maturation at growth cessation. Decrease BMD at lumbar spine.	[64,65,66]
**Androgen excess**			
Polycystic ovary syndrome (PCOS)	Hirsutism, acne, alopecia, seborrhea. Subfertility, menstrual dysfunction. Endometrial hyperplasia	No difference of BMD at the hip. No difference or lower spine BMD compared to healthy control in two studies. Decrease or increase fracture risk were both reported, may dependent on BMI.	[67,68,69,70]

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
