# Peer review of "Androgens and Androgen Receptor Actions on Bone Health and Disease: From Androgen Deficiency to Androgen Therapy"

_cells, 2019, doi:10.3390/cells8111318_

Round 1

Reviewer 1 Report

This review discusses the effects of androgens on bone cells.

The only thing these authors accomplish well, is to emphasize the importance of AR for mineralization. However the figures are a bit overconfident, proposing a mechanism based mainly on their own work.

Otherwise the manuscript's writing is mediocre, the manuscript is poorly structured and often confusing, and not written in an educational manner. There is a very broad discussion of a lot of endocrine disorders which is partly out of scope.

MAJOR COMMENTS:

1/ In lines 79-80, the authors state: "While still contradictory, most studies in rodents suggest that periosteal expansion is mediated via AR".

There is nothing contradictory about this, please rephrase. 

2/ The authors state: "This leads us to the hypothesis that periosteal bone formation can be accomplished via activation of AR pathway, which is inhibited by ER in excess of estrogens. This scenario might explain earlier bone growth arrest in pubertal girls than in boys."

Firstly, the use of "us" is misleading here; you did not came up with this idea yourself. Secondly, periosteal expansion has nothing to do with longitudinal growth arrest, so the second sentence is clearly erroneous.

3/ Section 2.1. on "bone growth" needs a thorough rewrite. First, you need to address longitudinal growth and how androgens influence that. Secondly, you need to address periosteal expansion. You need to integrate growth hormone/IGF1 imprinting by androgens and the results of the Callewaert 2010 paper JBMR.

4/ This statement needs to be revised: "The reported 93 mechanisms of androgen and/or estrogen effects on decreasing trabecular bone turnover may be 94 partly direct (bind to osteoclasts directly, induce osteoclasts apoptosis), but more likely indirect 95 (binding to bone marrow osteoblast precursors to inhibit osteoclastogenesis) [5,15]."

What are you talking about here: males? females?

5/ Section 2.3. not only has an unclear title, the contents are meaningless. Please provide an accurate account of the regulation of bone turnover and adult bone loss, specifying findings on male osteoporosis.

6/ The account of testosterone deficiency in aging is wrong and needs a thorough revision.

MINOR COMMENTS:

7/ In lines 51-53, the authors state "While some of the skeletal effects are primarily mediated via T itself (e.g., longitudinal bone growth), 51 other bone processes depend equally on T and DHT and/or their additive effects (e.g., periosteal 52 bone formation in radial growth, bone remodeling) [5]"

This statement is misleading for the unwary reader. The contribution of T vs DHT via AR is not really defined (and likely redundant). You are trying to say that some of the effects of T are mediated via ERs while others via AR. Please rephrase.

8/ Line 137: "decreased BMD at the spine, femur and distal"... Distal what?

9/ "particularly evident at bone compartments with large surface areas, such as cancellous bone" => read some work by Zebaze & Seeman, Lancet 2010 and you will be surprised how wrong this statement is.

10/ This statement is self-evident and poorly written: "Non-aromatized androgens do not appear to be as effective as aromatizable, maybe because they undergo aromatization through estrogens and stimulation of the ERs."

Author Response

Response to the reviewers’ comments

           Thanks to the editors and reviewers who identified areas of our manuscript that needed corrections or modifications. Here are our responses to improve the manuscript, in light of the reviewers' comments. (Point-by-point).

1/ In lines 79-80, the authors state: "While still contradictory, most studies in rodents suggest that periosteal expansion is mediated via AR". There is nothing contradictory about this, please rephrase. 

Thank you for the correction. We have rephrased this sentence in line 93~94, page 2 as “ most studies in rodents suggest that periosteal expansion is mediated via AR.”

2/ The authors state: "This leads us to the hypothesis that periosteal bone formation can be accomplished via activation of AR pathway, which is inhibited by ER in excess of estrogens. This scenario might explain earlier bone growth arrest in pubertal girls than in boys."

Firstly, the use of "us" is misleading here; you did not came up with this idea yourself. Secondly, periosteal expansion has nothing to do with longitudinal growth arrest, so the second sentence is clearly erroneous.

We appreciate the important opinion and correction pointed out by the reviewer. It is an incorrect description and we have corrected it in line 96~98, page 3 as ” This implicated that periosteal bone formation can be accomplished via activation of AR pathway.This scenario might explain earlier bone growth arrest in pubertal girls than in boys”, and we also delete the second sentence” which is inhibited by ER more than estrogens.”

3/ Section 2.1. on "bone growth" needs a thorough rewrite. First, you need to address longitudinal growth and how androgens influence that. Secondly, you need to address periosteal expansion. You need to integrate growth hormone/IGF1 imprinting by androgens and the results of the Callewaert 2010 paper JBMR.

It is grateful to provide the important suggestion. We have rewritten this paragraph of section 2.1 in line 60~83, 101~112, page 2~3 as the following:

“Androgens are known to stimulate longitudinal bone growth as well as radial bone growth, thereby increasing the cortical bone size. Longitudinal bone grows through endochondral bone formation and epiphyseal plate growth, whereas radial bone grows through periosteal apposition. Cartilage cells, predominantly chondrocytes, proliferate and differentiate under the regulation of various endocrines, such as growth hormone (GH), insulin-like growth factor-I (IGF-I), transforming growth factor (TGE-beta), and vitamin D metabolites [6]. Longitudinal bone growth is governed by the sex hormones which exerts biphasic effects during adolescence: As puberty begins, androgen and estrogen stimulate endochondral bone development, but the epiphyseal growth plate closure is majorly mediated by estrogen via ER at the end of puberty through aromatization of androgen to estrogen [7]. Estrogen demonstrates biphasic effects on epiphyseal growth, where low concentration, as male sex physiologically presented, can stimulate epiphyseal growth, whereas higher concentration level, as female sex presented, is associated with arrest of bone growth [8]. This is testified by the observed growth spurt in puberty due to delayed closure of epiphyses in estrogen-deficient individuals (e.g., mutated aromatase gene) or estrogen-resistant cases (ER gene mutation) [9,10].

A greater radial bone expansion, comprising enlargement of bone diameter and increase of cortical thickness, is characteristic for bone growth in male puberty. Accelerated periosteal bone apposition in men than in women [11]is conventionally supposed to result from stimulatory effects of androgens on periosteal bone through AR in men and inhibitory effects of estrogens in women during puberty. Experiments in transgenic mice provided evidence that estrogen may also stimulate radial bone growth. Observational studies on men with aromatase deficiency, characterized by normal serum androgen but undetectable level or estrogen, exhibiting low bone mass and areal density. Take these findings together, periosteal bone expended is not only mediated by androgen, but also estrogen.“

“GH/ IGF-1 axis is essential in achieving a normal longitudinal bone growth and bone mass acquisition in association of sex steroids.GH not only directly promotes differentiation of mesenchymal stem cells (MSC) into osteoblasts but stimulate osteoblast proliferation[16,17]. GH may promote osteoblast differentiation indirectly by upregulating of bone morphogenetic proteins (BMPs) and IGF-1/IGF-2B[18]. IGF-1, majorly produced in the liver, exerts stimulatory effects though binding IGF-1 receptors on osteoblast resulting in cell differentiation[19], and the serum level is positively correlated with BMD in older women and men[20]. Circulating estrogen, whether endogenous or exogenous origin, may modulate GH/IGF-1 axis, influencing bone turnover. In a GH receptor gene knockout (GHRKO) mice study, GH/IGF-I activation is not involved in the development and maintenance of trabecular bone, while androgens stimulate trabecular bone modeling independently from GH/IGH-1. However, GH/IGF-1 activation are the major determinants of periosteal bone growth to obtain an optimal periosteal bone growth[13]. “

4/ This statement needs to be revised: "The reported mechanisms of androgen and/or estrogen effects on decreasing trabecular bone turnover may be partly direct bind to osteoclasts directly, induce osteoclasts apoptosis, but more likely indirect binding to bone marrow osteoblast precursors to inhibit osteoclastogenesis [5,15]." What are you talking about here: males? females?

Thank you for pointing out this question. This description is based on the animal study, using male transgenic mice. We have corrected this sentence in line 118~121, page 3 as “The reported mechanisms of androgen effects on trabecular bone turnover may directly bind to osteoclasts, inducing osteoclasts apoptosis, or indirectly interacts with osteoblast precursors, inhibiting osteoclastogenesis. ”

5/ Section 2.3. not only has an unclear title, the contents are meaningless. Please provide an accurate account of the regulation of bone turnover and adult bone loss, specifying findings on male osteoporosis.

We thank the comment from reviewer. We have revised the whole section in page 4, line 135-152 as following:

“ 2.3 Regulation of bone turnover in men.

Bone turnover markers (BTM), products derived from bone matrix or bone cells during bone remodeling, could be categorized by either catabolic (bone resorption, such as CTx, NTx, pyridinolines) or anabolic markers (bone formation, such as procollagen type I N-terminal propeptide [PINP], bone alkaline phosphatase [bone ALP] or osteocalcin [OC]). For men, the bone turnover markers are highest at the second or third decade, then mildly decreases hereafter. In young man, the relationship between increased bone turnover and high levels of sex hormone or IGF-1 is stronger than that in old men[25]. Data from Dubbo, MINOS and MrOS cohort are unable to prove conventional BTM as an independent biomarker to predict fragility fracture in older men[26-28].

AR can be found on osteoblasts, osteoclasts and osteocytes, but mainly expressed on the osteoblast[29], to a lesser degree, on the bone marrow cell regulating osteoclastogenesis[30]. Regulation of osteoclast function is primary mediated by Estrogen and ER, and T may control osteoclast indirectly by aromatization to E2, or by inhibiting the production of interleukin-6, which essential for the maturation of osteoclast precursors[30]. AR in osteoblast is up-regulated by androgen, estrogen and 1,25 hydroxy D3 to stimulate osteoblast proliferation, differentiation and synthesis of extracellular matrix protein and mineralization[31,32]. AR in osteocytes had a direct role in maintaining the skeletal integrity and bone quality[33]. However, AR had no role of direct action on osteoclast in animal model study[34].“

6/ The account of testosterone deficiency in aging is wrong and needs a thorough revision.

Thank you for your comments and we have revised this paragraphin line 264~279, page 7as following:

“Aging in men results in a progressive reduction of hypothalamic-pituitary-gonadal (HPG) axis function, decreasing testosterone secretion through both central and peripheral origin. In men, serum total T declined with advancing age in a rate of 1~2% per year since the 3rd decade[107]. At the age of 75 y, the circulating T level may loss 30% of which in a 25-y/o young man [108]. Data from Osteoporotic Fractures in Men Study (MrOS) indicated that sex hormone deficiency was associated with higher prevalence of osteoporosis at baseline and greater loss of BMD over time in old age menaccompanied with parallel T and E2 decline, of which 3% were Tdeficient, 3.2% were E2deficient and 0.7% were deficient in both. In old age men, there was a threshold level of E2for fracture [109]but no association between total T and fracture, implicating that E2, but not T, may be the major sex hormone associated with fracture risk in older men. On the contrary, some studies indicated that serum T is associated with fractures independently[41]and more powerfully than E2[110]. For instance, high bone resorption in men who sustained hip fracture correlated well with low serum T levels[111]. However, old age and sex hormone deficiency might be two distinct risk factors of cortical bone lossin old age from an animal study, which addressed the major mechanism of estrogen deficiency on bone through increase osteoclastogenesis; whereas aging was through decrease osteoblastogenesis in combination with increase osteoclastogeneisis [112].“

MINOR COMMENTS:

7/ In lines 51-53, the authors state "While some of the skeletal effects are primarily mediated via T itself (e.g., longitudinal bone growth), 51 other bone processes depend equally on T and DHT and/or their additive effects (e.g., periosteal 52 bone formation in radial growth, bone remodeling) [5]". This statement is misleading for the unwary reader. The contribution of T vs DHT via AR is not really defined (and likely redundant). You are trying to say that some of the effects of T are mediated via ERs while others via AR. Please rephrase.

We appreciate reviewer’s comment. The original sentence is complex and we rephase it in line 50~56, page 2 as following:

“Activation of AR and ERα, but not ERβ, is associated with maintenance of trabecular bone. Effects of ERα activation preserved the thickness and number in trabeculae, while AR preserved the number of trabeculae[5]. For cortical bone, the bone-sparing effects aremajorly mediated by ERα, but not AR or ERβ for the maintenance of the cortical thickness, volumetric density and mechanical strength[5]. Activation of ERα may have direct impacts on bone, or indirectly through increasing IGF-1 in serum[5].“

8/ Line 137: "decreased BMD at the spine, femur and distal"... Distal what?

We thank the question raised by the reviewer. We intended to indicate distal site of non-dominant radial bone, and we have corrected this sentence in line 163~164, page 4 as “Impaired skeletal growth with lower peak bone mass achieved reflects in decreased BMD at the spine, femur and distal sites of non-dominant radial bone.”

9/ "particularly evident at bone compartments with large surface areas, such as cancellous bone" => read some work by Zebaze & Seeman, Lancet 2010 and you will be surprised how wrong this statement is.

We appreciate the reviewer to point out this question. We have corrected this paragraph in line 254~258, page 7 as following:

“Based on these findings, there is much resemblance in bone deterioration between androgen-deficient men and postmenopausal women or hypogonadal animals. In this sense, imbalances in bone remodeling with a shift to resorption would cause net bone loss, which is evident at intracortical or endocortical surface in the appendicular skeleton and vertebral bone[102,103].”

10/ This statement is self-evident and poorly written: "Non-aromatized androgens do not appear to be as effective as aromatizable, maybe because they undergo aromatization through estrogens and stimulation of the ERs."

We thank the reviewer to indicate this problem. we have clarified this sentence and have rewritten it in line 363~367, page 9, as following:

“ DHT was able to reverse the loss of cancellous bone but DHT did not prevent cortical thinning induced by ORX[28]. In summary, both DHT and T in OVX female and ORX male rats were protective for bone integrity, in particular in the cancellous bone. DHT do not appear to be as effective as T in cortical bones , maybe because T can undergo aromatization through estrogens and stimulation of the ERs.”

Reviewer 2 Report

This work by Jia-Feng Chen and colleagues constitutes a very interesting and comprehensive review on the role of androgens and androgen receptors (ARs) on bone health and disease. The manuscript adequately describes the main actions exerted in bone by androgen and ARs in androgen deficiency disease and in androgen therapy both in animal models and in human.

I believe that the manuscript can be more usable by adding some diagrams/schemes that reflect the main topics of the review; this would make it much easier for a reader to follow.

Furthermore, the structure of the manuscript would improve if the authors dedicate a separate subsection (maybe in section 2) to the effect of androgens on bone cells (osteoblasts, osteoclasts...).

Figure 1 is too small and should be ameliorated in terms of quality and size.

Some minor syntax and language errors need to be corrected.

Author Response

Response to the reviewers’ comments

           Thanks to the editors and reviewers who identified areas of our manuscript that needed corrections or modifications. Here are our responses to improve the manuscript, in light of the reviewer’s comments. (Point-by-point).

1. I believe that the manuscript can be more usable by adding some diagrams/schemes that reflect the main topics of the review; this would make it much easier for a reader to follow.

We are grateful for the suggestion from reviewer. A summarized Table 1 that reflects the main topics of the reviewhas been added in page 5 to describe the disease-specific impacts on bone.

2. Furthermore, the structure of the manuscript would improve if the authors dedicate a separate subsection (maybe in section 2) to the effect of androgens on bone cells (osteoblasts, osteoclasts...).

Thank you for your suggestion. A paragraph regarding the effects of androgens on bone cells has been added in line 135~152, page 3-4 as following:

“Bone turnover markers (BTM), products derived from bone matrix or bone cells during bone remodeling, could be categorized by either catabolic (bone resorption, such as CTx, NTx, pyridinolines) or anabolic markers (bone formation, such as procollagen type I N-terminal propeptide [PINP], bone alkaline phosphatase [bone ALP] or osteocalcin [OC]). For men, the bone turnover markers are highest at the second or third decade, then mildly decreases hereafter. In young man, the relationship between increased bone turnover and high levels of sex hormone or IGF-1 is stronger than that in old men[25]. Data from Dubbo, MINOS and MrOS cohort are unable to prove conventional BTM as an independent biomarker to predict fragility fracture in older men[26-28].

AR can be found on osteoblasts, osteoclasts and osteocytes, but mainly expressed on the osteoblast[29], to a lesser degree, on the bone marrow cell regulating osteoclastogenesis[30]. Regulation of osteoclast function is primary mediated by Estrogen and ER, and T may control osteoclast indirectly by aromatization to E2, or by inhibiting the production of interleukin-6, which essential for the maturation of osteoclast precursors[30]. AR in osteoblast is up-regulated by androgen, estrogen and 1,25 hydroxy D3 to stimulate osteoblast proliferation, differentiation and synthesis of extracellular matrix protein and mineralization[31,32]. AR in osteocytes had a direct role in maintaining the skeletal integrity and bone quality[33]. However, AR had no role of direct action on osteoclast in animal model study[34].”  

3. Figure 1 is too small and should be ameliorated in terms of quality and size.

Thank you for your suggestion. The quality of figures has been improved as reviewer suggested by enlarging the figure size and ameliorating the picture quality in page 11.

4. Some minor syntax and language errors need to be corrected.

Thank you for your suggestion. Some syntax and language errors have been corrected. We marked the corrected sentence as yellow background in the text.

Reviewer 3 Report

The review by Chen et al. summarizes the role of androgens in bone homeostasis, development and bone mineral density in various aspects including the distinction by gender, interplay with estrogens, hypogonadism, Klinefelter syndrome, AIS - Reifenstein syndrome, androgen deprivation therapy and androgen receptor antagonist treatments as well as decline of androgens by aging. Authors discuss also underlying molecular mechanisms of androgen signalling in bone, androgen therapy as well as suppression of adipogenesis.

This is a very comprehensive and excellent review of this topic that has not been summarized in this broad manner.

This review provides a very good overview of this important up-to-date topic.

Minor comments:

Fig. 1 will be hard to see at this small size. Please revise Fig. 1.

Author Response

Response to the reviewers’ comments

           Thanks to the editors and reviewers who identified areas of our manuscript that needed corrections or modifications. Here are our responses to improve the manuscript, in light of the reviewer’s comments. (Point-by-point).

Fig. 1 will be hard to see at this small size. Please revise Fig. 1.

Thanks for the suggestion from the reviewer, we have revised Figure 1 with larger size and higher resolution on page 11.

Reviewer 4 Report

Hong-Yo Kang and colleagues have submitted a review article on androgens and androgen receptor (AR) actions on bone in health and disease. The topic of the review is quite interesting and the authors have covered a vast area of androgen action. At the end they have concentrated on the molecular action of AR in skeletal stem and progenitor cells. Because of the fairly broad nature of the areas covered, different authors from different expertise seem to have written the article. The end result, unfortunately, is an inhomogeneous piece of article which does not make easy reading because of differences in the style of writing.

While some sections are well written, other sections are poorly written and have language problems. Some of the sections also assume that the readers are in their research fields and have used acronyms and abbreviations which could only be understood by aficionados in the field.

Sentences such as:

Line 41: “Androgens bind to AR, which translocates into the nucleus….” translocates from where?

Line 239: “For instance, high bone resorption in men who sustained s hip fracture correlated well with low serum T level“  What is s hip fracture?

Line 246: “…failed to find any no consistent relationship between serum DHEA-S and BMD [20,91].  It is not clear what this means.

Line 275-276: “Since a decreased BMD can be found in some women with intact gonads, it was suggested that supplementing with additional estrogens could be helpful for them even if they demonstrate intact testes [106].” What does this mean? Women demonstrating intact testes??

Line 152: “..and serum bone markers[42] implied androgen is contribution to bone formation”.  What does this mean?

Line 144: “T supplement is also proved to be contributed to BMD incensement in men with low testosterone..”  What does this mean?   

These are a few of the many statements that are incorrectly presented (in content and language) that make is very difficult to follow this article.

Author Response

Response to the reviewers’ comments

           Thanks to the editors and reviewers who identified areas of our manuscript that needed corrections or modifications. Here are our responses to improve the manuscript, in light of the reviewer’s comments. (Point-by-point).

While some sections are well written, other sections are poorly written and have language problems. Some of the sections also assume that the readers are in their research fields and have used acronyms and abbreviations which could only be understood by aficionados in the field. Sentences such as: Line 41: “Androgens bind to AR, which translocates into the nucleus….” translocates from where?

We are grateful for the comments. The binding of androgen to AR occurred in the cytoplasm, then translocates to the nucleus. We have clarified the sentence in line 41~42, page 1 as” Androgens bind to AR in the cytoplasm, translocating into the nucleus and binding to DNA to function as a transcription factor.”

Line 239: “For instance, high bone resorption in men who sustained s hip fracture correlated well with low serum T level“ What is s hip fracture?

 We appreciate the correction by the reviewer. “s hip fracture” is a clerical error, and we have rewritten this sentence in line 275~276, page 7 as ” For instance, high bone resorption in men who sustained hip fracture correlated well with low serum T levels[111].”

Line 246: “…failed to find any no consistent relationship between serum DHEA-S and BMD [20,91].  It is not clear what this means.

We thank the reviewer for pointing-out the mistake, and we have specified this sentence on line 282~284,page 7 as “While there were no prospective studies on the relationship with fracture risk, previous cross-sectional studies failed to find any consistent relationship between serum DHEA-S and BMD.”

Line 275-276: “Since a decreased BMD can be found in some women with intact gonads, it was suggested that supplementing with additional estrogens could be helpful for them even if they demonstrate intact testes [106].” What does this mean? Women demonstrating intact testes??

 We thank the reviewer to point-out the error. Actually, individual with complete androgen insensitivity syndrome are born phenotypically female, without any signs of genital masculinization, despite of having a 46XY karyotype. Thus, this sentence is incorrect, and we have rewritten it in line 312~314, page 8 as “Since a decreased BMD can be found in some cases with intact gonads, it was suggested that supplementing with additional estrogens could be helpful for them even if they demonstrate intact testes.”

Line 152: “..and serum bone markers[42] implied androgen is contribution to bone formation”.  What does this mean?

 We appreciate the reviewer to point out the question. Our intention is to indicate that testosterone therapy is correlated well with the absorption of 1,25(OH)2D/ calcium and bone formation marker, and hence androgen might be contributory to bone formation. Therefore, we have clarified this sentence in line 179-181, page 4 as” T supplementary therapy significantly increases in 1,25(OH)2D/ calcium absorption [40] and serum bone formation markers[42], implicating that androgen is contributory to bone formation.” 

Line 144: “T supplement is also proved to be contributed to BMD incensement in men with low testosterone.”  What does this mean?   

Thanks for the question from the reviewer. We have clarified this sentence in line 172-173, page 4 as ” T supplementary therapy has also been proved to contribute to BMD increment in men with low testosterone levels in several studies.“

These are a few of the many statements that are incorrectly presented (in content and language) that make is very difficult to follow this article.

Thank you for your correction. Some incorrect syntax and language errors in the main text have been corrected, and we marked the corrected sentence in yellow background.

Round 2

Reviewer 1 Report

I would like to thank the authors for their efforts to improve the manuscript.

Several points are still insufficiently addressed (numbering refers to previous comments)

1/ Periosteal expansion IS mediated via AR (you can leave out the "most studies"). However, the target cell(s) mediating this effect of AR remain unclear.

2/ "This implicated that periosteal bone formation can be accomplished via activation of AR pathway.This scenario might explain earlier bone growth arrest in pubertal girls than in boys”

=> you are still mixing up "growth arrest" (=longitudinal growth) with periosteal expansion. The earlier growth arrest in girls is due to earlier onset of puberty. Refer to Almeida et al. Physiol Rev 2017 for a detailed analysis of longitudinal growth and sex steroids.

4/ This explanation is wrong: line 118~121, page 3 as “The reported mechanisms of androgen effects on trabecular bone turnover may directly bind to osteoclasts, inducing osteoclasts apoptosis, or indirectly interacts with osteoblast precursors, inhibiting osteoclastogenesis. ”

=> Sinnesael et al. as well as Ucer et al. have shown that AR plays absolutelety no role in osteoclasts!

5/ This sentence is wrong: "AR can be found on osteoblasts, osteoclasts and osteocytes, but mainly expressed on the osteoblast[29], to a lesser degree, on the bone marrow cell regulating osteoclastogenesis[30]."

=> Refer to Sinnesael et al., Mol Cell Endocrinol, and Sinnesael et al., JBMR who demonstrated that AR is expressed in osteeoblasts and even more so in ostocytes, but not in osteoclasts. Other studies are likely flawed and based on non-specific primary antibodies.

6/ The account of testosterone deficiency in aging (line 264~279, page 7) is still missing an important discussion on the role of SHBG and free testosterone. 

Refer to:

- Travison TG et al. J Clin Endocrinol Metab. 2017 Apr 1;102(4):1161-1173.

- Bhasin S et al. J Clin Endocrinol Metab. 2011 Aug;96(8):2430-9.

Author Response

We would like to take this opportunity to express our sincere thanks to the editors and reviewers who identified areas of our manuscript that needed corrections or modifications. We also would like to thank you for recognizing and bringing out the best in our research findings.

Response to the reviewers’ comments

            Here are our responses to improve the manuscript, in light of the reviewer’s comments. (Point-by-point).

1/ Periosteal expansion is mediated via AR (you can leave out the "most studies"). However, the target cell(s) mediating this effect of AR remain unclear.

Reply: Thank you for the correction, and we have rephrased this sentence as the reviewer’s suggestion on page 2, line 93~94 as” Periosteal expansion is mediated via AR and ERα pathway[15]. However, the target cell(s) mediating this effects remain unclear.”

2/ "This implicated that periosteal bone formation can be accomplished via activation of AR pathway. This scenario might explain earlier bone growth arrest in pubertal girls than in boys”

=> you are still mixing up "growth arrest" (=longitudinal growth) with periosteal expansion. The earlier growth arrest in girls is due to earlier onset of puberty. Refer to Almeida et al. Physiol Rev 2017for a detailed analysis of longitudinal growth and sex steroids.

Reply: We appreciate the reviewer to point out this mistake. The bone growth arrest has nothing to do with periosteal bone formation, as the provided paper indicated. The second sentence was incorrect and we have deleted it. It has been rephrased on page 3, line 96~99 as ” The longitudinal bone growth is facilitated by chondrocytes at the epiphyseal plates, closure of which is mediated by estrogens on the chondrocytes[15]. This scenario might explain earlier growth arrest is attributed to the earlier onset of puberty in female.”

4/ This explanation is wrong: line 118~121, page 3 as “The reported mechanisms of androgen effects on trabecular bone turnover may directly bind to osteoclasts, inducing osteoclasts apoptosis, or indirectly interacts with osteoblast precursors, inhibiting osteoclastogenesis. ”

=> Sinnesael et al. as well as Ucer et al. have shown that AR plays absolutely no role in osteoclasts!

Reply: We thank the reviewer to provide the important reference data on the role of AR on osteoclast. The original sentence was inappropriate and we have rephrased it on page 3, line 119~122 as “ Under the androgen effects, the trabecular bone may be maintained directly by osteocytes, or indirectly by inhibiting osteoclastogenesis through interaction with osteoblast precursors, while AR has no role on the osteoclasts[23-26].”we also have revised this sentence on page 4, line 145~149 as following: “AR is predominantly expressed on osteoblasts and osteocytes, but not on osteoclasts[24,32,33]. It has been shown that osteoclast-specific ARKO mice have no changes neither in osteoclast surface nor in bone microarchitecture nor in the response to orchidectomy and androgen replacement, suggesting that the AR in osteoclasts has no critical role for bone maintenance in mice [32].”

5/ This sentence is wrong: "AR can be found on osteoblasts, osteoclasts and osteocytes, but mainly expressed on the osteoblast[ ], to a lesser degree, on the bone marrow cell regulating osteoclastogenesis[30]."

=> Refer to Sinnesael et al., Mol Cell Endocrinol, and Sinnesael et al., JBMR who demonstrated that AR is expressed in osteoblasts and even more so in osteocytes, but not in osteoclasts. Other studies are likely flawed and based on non-specific primary antibodies.

Reply: We thank the reviewer to provide this important information, we have revised this sentence on page 4, line 145~149 as following: “AR is predominantly expressed on osteoblasts and osteocytes, but not on osteoclasts[24,32,33]. It has been shown that osteoclast-specific ARKO mice have no changes neither in osteoclast surface nor in bone microarchitecture nor in the response to orchidectomy and androgen replacement, suggesting that the AR in osteoclasts has no critical role for bone maintenance in mice [32].”

6/ The account of testosterone deficiency in aging (line 264~279, page 7) is still missing an important discussion on the role of SHBG and free testosterone. 

Refer to:

- Travison TG et al. J Clin Endocrinol Metab. 2017 Apr 1;102(4):1161-1173.

- Bhasin S et al. J Clin Endocrinol Metab. 2011 Aug;96(8):2430-9.

Reply: We highly appreciate this excellent suggestion from the reviewer. We have rewritten this paragraph and added the discussion about the SHBG and free testosterone on page 7, line 262~285 as following:

“ Circulating T is transported predominantly by sex hormone binding globulin (SHBG), which controls the amount of T in the body. SHBG-bound T is biologically inactive due to high affinity, whereas free T (approximately 2~3%) [109] and albumin-bound T are considered bioavailable, correlated better with BMD and muscle mass than total T[110,111].  In aging men, total T declined at the speed of 1.6% per year, while 2~3% per year for bioavailable T [112].  However, SHGB level increases with aging, resulting in the greater reduction of bioavailable T level[112]. Serum total and free T, inversely correlated with age, are associated with sexual symptoms, physical activity and metabolic conditions in different cohorts[113-115]. The reference range of T, influenced by races, regions, timing of sample collection and laboratory methods, needed to be harmonized to avoid inter-cohort variation[116].

Aging in men results in a progressive reduction of hypothalamic-pituitary-gonadal (HPG) axis function, decreasing testosterone secretion through both central and peripheral origin. Data from Osteoporotic Fractures in Men Study (MrOS) indicated that sex hormone deficiency was associated with higher prevalence of osteoporosis at baseline and greater loss of BMD over time in old age men accompanied with parallel T and E2 decline, of which 3% were T deficient, 3.2% were E2 deficient and 0.7% were deficient in both. In old age men, there was a threshold level of E2 for fracture [117] but no association between total T and fracture, implicating that E2, but not T, may be the major sex hormone associated with fracture risk in older men. On the contrary, some studies indicated that serum T is associated with fractures independently[43] and more powerfully than E2[118]. For instance, high bone resorption in men who sustained hip fracture correlated well with low serum T levels[119]. However, old age and sex hormone deficiency might be two distinct risk factors of cortical bone loss in old age from an animal study, which addressed the major mechanism of estrogen deficiency on bone through increase osteoclastogenesis; whereas aging was through decrease osteoblastogenesis in combination with increase osteoclastogeneisis [120]. “